# Probabilistic Principal Geodesic Analysis

**Miaomiao Zhang**
School of Computing
University of Utah
Salt Lake City, UT
miaomiao@sci.utah.edu

**P. Thomas Fletcher**
School of Computing
University of Utah
Salt Lake City, UT
fletcher@sci.utah.edu

## Abstract

Principal geodesic analysis (PGA) is a generalization of principal component analysis (PCA) for dimensionality reduction of data on a Riemannian manifold. Currently PGA is defined as a geometric fit to the data, rather than as a probabilistic model. Inspired by probabilistic PCA, we present a latent variable model for PGA that provides a probabilistic framework for factor analysis on manifolds. To compute maximum likelihood estimates of the parameters in our model, we develop a Monte Carlo Expectation Maximization algorithm, where the expectation is approximated by Hamiltonian Monte Carlo sampling of the latent variables. We demonstrate the ability of our method to recover the ground truth parameters in simulated sphere data, as well as its effectiveness in analyzing shape variability of a corpus callosum data set from human brain images.

## 1 Introduction

Principal component analysis (PCA) [12] has been widely used to analyze high-dimensional data. Tipping and Bishop proposed probabilistic PCA (PPCA) [22], which is a latent variable model for PCA. A similar formulation was proposed by Roweis [18]. Their work opened up the possibility for probabilistic interpretations for different kinds of factor analyses. For instance, Bayesian PCA [3] extended PPCA by adding a prior on the factors, resulting in automatic selection of model dimensionality. Other examples of latent variable models include probabilistic canonical correlation analysis (CCA) [1] and Gaussian process latent variable models [15]. Such latent variable models have not, however, been extended to handle data from a Riemannian manifold.

Manifolds arise naturally as the appropriate representations for data that have smooth constraints. For example, when analyzing directional data [16], i.e., vectors of unit length in $\mathbb{R}^n$, the correct representation is the sphere, $S^{n-1}$. Another important example of manifold data is in shape analysis, where the definition of the shape of an object should not depend on its position, orientation, or scale. Kendall [14] was the first to formulate a mathematically precise definition of shape as equivalence classes of all translations, rotations, and scalings of point sets. The result is a manifold representation of shape, or *shape space*. Linear operations violate the natural constraints of manifold data, e.g., a linear average of data on a sphere results in a vector that does not have unit length. As shown recently [5], using the kernel trick with a Gaussian kernel maps data onto a Hilbert sphere, and utilizing Riemannian distances on this sphere rather than Euclidean distances improves clustering and classification performance. Other examples of manifold data include geometric transformations, such as rotations and affine transforms, symmetric positive-definite tensors [9, 24], Grassmannian manifolds (the set of $m$-dimensional linear subspaces of $\mathbb{R}^n$), and Stiefel manifolds (the set of orthonormal $m$-frames in $\mathbb{R}^n$) [23]. There has been some work on density estimation on Riemannian manifolds. For example, there is a wealth of literature on parametric density estimation for directional data [16], e.g., spheres, projective spaces, etc. Nonparametric density estimation based on kernel mixture models [2] was proposed for compact Riemannian manifolds. Methods for sampling from manifold-valued distributions have also been proposed [4, 25]. It's important to note

the distinction between manifold data, where the manifold representation is known *a priori*, versus manifold learning and nonlinear component analysis [15, 20], where the data lies in Euclidean space on some unknown, lower-dimensional manifold that must be learned.

Principal geodesic analysis (PGA) [10] generalizes PCA to nonlinear manifolds. It describes the geometric variability of manifold data by finding lower-dimensional geodesic subspaces that minimize the residual sum-of-squared geodesic distances to the data. While [10] originally proposed an approximate estimation procedure for PGA, recent contributions [19, 21] have developed algorithms for exact solutions to PGA. Related work on manifold component analysis has introduced variants of PGA. This includes relaxing the constraint that geodesics pass through the mean of the data [11] and, for spherical data, replacing geodesic subspaces with nested spheres of arbitrary radius [13]. All of these methods are based on geometric, least-squares estimation procedures, i.e., they find subspaces that minimize the sum-of-squared geodesic distances to the data. Much like the original formulation of PCA, current component analysis methods on manifolds lack a probabilistic interpretation. In this paper, we propose a latent variable model for PGA, called probabilistic PGA (PPGA). The model definition applies to generic manifolds. However, due to the lack of an explicit formulation for the normalizing constant, our estimation is limited to *symmetric spaces*, which include many common manifolds such as Euclidean space, spheres, Kendall shape spaces, Grassman/Stiefel manifolds, and more. Analogous to PPCA, our method recovers low-dimensional factors as maximum likelihood.

## 2  Riemannian Geometry Background

In this section we briefly review some necessary facts about Riemannian geometry (see [6] for more details). Recall that a Riemannian manifold is a differentiable manifold $M$ equipped with a metric $g$, which is a smoothly varying inner product on the tangent spaces of $M$. Given two vector fields $v, w$ on $M$, the covariant derivative $\nabla_v w$ gives the change of the vector field $w$ in the $v$ direction. The covariant derivative is a generalization of the Euclidean directional derivative to the manifold setting. Consider a curve $\gamma : [0, 1] \to M$ and let $\dot{\gamma} = d\gamma/dt$ be its velocity. Given a vector field $V(t)$ defined along $\gamma$, we can define the covariant derivative of $V$ to be $\frac{DV}{dt} = \nabla_{\dot{\gamma}} V$. A vector field is called parallel if the covariant derivative along the curve $\gamma$ is zero. A curve $\gamma$ is geodesic if it satisfies the equation $\nabla_{\dot{\gamma}} \dot{\gamma} = 0$. In other words, geodesics are curves with zero acceleration.

Recall that for any point $p \in M$ and tangent vector $v \in T_p M$, the tangent space of $M$ at $p$, there is a unique geodesic curve $\gamma$, with initial conditions $\gamma(0) = p$ and $\dot{\gamma}(0) = v$. This geodesic is only guaranteed to exist locally. When $\gamma$ is defined over the interval $[0, 1]$, the Riemannian exponential map at $p$ is defined as $\mathrm{Exp}_p(v) = \gamma(1)$. In other words, the exponential map takes a position and velocity as input and returns the point at time 1 along the geodesic with these initial conditions. The exponential map is locally diffeomorphic onto a neighbourhood of $p$. Let $V(p)$ be the largest such neighbourhood. Then within $V(p)$ the exponential map has an inverse, the Riemannian log map, $\mathrm{Log}_p : V(p) \to T_p M$. For any point $q \in V(p)$, the Riemannian distance function is given by $d(p, q) = \| \mathrm{Log}_p(q) \|$. It will be convenient to include the point $p$ as a parameter in the exponential and log maps, i.e., define $\mathrm{Exp}(p, v) = \mathrm{Exp}_p(v)$ and $\mathrm{Log}(p, q) = \mathrm{Log}_p(q)$. The gradient of the squared distance function is $\nabla_p d(p, q)^2 = -2 \mathrm{Log}(p, q)$.

## 3  Probabilistic Principal Geodesic Analysis

Before introducing our PPGA model for manifold data, we first review PPCA. The main idea of PPCA is to model an $n$-dimensional Euclidean random variable $y$ as

$$y = \mu + Bx + \epsilon, \tag{1}$$

where $\mu$ is the mean of $y$, $x$ is a $q$-dimensional latent variable, with $x \sim N(0, I)$, $B$ is an $n \times q$ factor matrix that relates $x$ and $y$, and $\epsilon \sim N(0, \sigma^2 I)$ represents error. We will find it convenient to model the factors as $B = W\Lambda$, where the columns of $W$ are mutually orthogonal, and $\Lambda$ is a diagonal matrix of scale factors. This removes the rotation ambiguity of the latent factors and makes them analogous to the eigenvectors and eigenvalues of standard PCA (there is still of course an ambiguity of the ordering of the factors). We now generalize this model to random variables on Riemannian manifolds.

## 3.1 Probability Model

Following [8, 17], we use a generalization of the normal distribution for a Riemannian manifold as our noise model. Consider a random variable $y$ taking values on a Riemannian manifold $M$, defined by the probability density function (pdf)

$$p(y|\mu, \tau) = \frac{1}{C(\mu, \tau)} \exp\left(-\frac{\tau}{2} d(\mu, y)^2\right),$$

$$C(\mu, \tau) = \int_M \exp\left(-\frac{\tau}{2} d(\mu, y)^2\right) dy. \tag{2}$$

We term this distribution a *Riemannian normal distribution*, and use the notation $y \sim N_M(\mu, \tau^{-1})$ to denote it. The parameter $\mu \in M$ acts as a location parameter on the manifold, and the parameter $\tau \in \mathbb{R}_+$ acts as a dispersion parameter, similar to the precision of a Gaussian. This distribution has the advantages that (1) it is applicable to any Riemannian manifold, (2) it reduces to a multivariate normal distribution (with isotropic covariance) when $M = \mathbb{R}^n$, and (3) much like the Euclidean normal distribution, maximum-likelihood estimation of parameters gives rise to least-squares methods (see [8] for details). We note that this noise model could be replaced with a different distribution, perhaps specific to the type of manifold or application, and the inference procedure presented in the next section could be modified accordingly.

The PPGA model for a random variable $y$ on a smooth Riemannian manifold $M$ is

$$y|x \sim N_M\left(\text{Exp}(\mu, z), \tau^{-1}\right), z = W\Lambda x, \tag{3}$$

where $x \sim N(0, 1)$ are again latent random variables in $\mathbb{R}^q$, $\mu$ here is a base point on $M$, $W$ is a matrix with $q$ columns of mutually orthogonal tangent vectors in $T_\mu M$, $\Lambda$ is a $q \times q$ diagonal matrix of scale factors for the columns of $W$, and $\tau$ is a scale parameter for the noise. In this model, a linear combination of $W\Lambda$ and the latent variables $x$ forms a new tangent vector $z \in T_\mu M$. Next, the exponential map shoots the base point $\mu$ by $z$ to generate the location parameter of a *Riemannian normal distribution*, from which the data point $y$ is drawn. Note that in Euclidean space, the exponential map is an addition operation, $\text{Exp}(\mu, z) = \mu + z$. Thus, our model coincides with (1), the standard PPCA model, when $M = \mathbb{R}^n$.

## 3.2 Inference

We develop a maximum likelihood procedure to estimate the parameters $\theta = (\mu, W, \Lambda, \tau)$ of the PPGA model defined in (3). Given observed data $y_i \in \{y_1, ..., y_N\}$ on $M$, with associated latent variable $x_i \in \mathbb{R}^q$, and $z_i = W\Lambda x_i$, we formulate an expectation maximization (EM) algorithm. Since the expectation step over the latent variables does not yield a closed-form solution, we develop a Hamiltonian Monte Carlo (HMC) method to sample $x_i$ from the posterior $p(x|y; \theta)$, the log of which is given by

$$\log \prod_{i=1}^N p(x_i|y_i; \theta) \propto -N \log C - \sum_{i=1}^N \frac{\tau}{2} d\left(\text{Exp}(\mu, z_i), y_i\right)^2 - \frac{\|x_i\|^2}{2}, \tag{4}$$

and use this in a Monte Carlo Expectation Maximization (MCEM) scheme to estimate $\theta$. The procedure contains two main steps:

### 3.2.1 E-step: HMC

For each $x_i$, we draw a sample of size $S$ from the posterior distribution (4) using HMC with the current estimated parameters $\theta^k$. Denote $x_{ij}$ as the $j$th sample for $x_i$, the Monte Carlo approximation of the $Q$ function is given by

$$Q(\theta|\theta^k) = E_{x_i|y_i; \theta^k} \left[\prod_{i=1}^N \log p(x_i|y_i; \theta^k)\right] \approx \frac{1}{S} \sum_{j=1}^S \sum_{i=1}^N \log p(x_{ij}|y_i; \theta^k). \tag{5}$$

In our HMC sampling procedure, the potential energy of the Hamiltonian $H(x_i, m) = U(x_i) + V(m)$ is defined as $U(x_i) = -\log p(x_i|y_i; \theta)$, and the kinetic energy $V(m)$ is a typical isotropic

Gaussian distribution on a $q$-dimensional auxiliary momentum variable, $m$. This gives us a Hamiltonian system to integrate: $\frac{dx_i}{dt} = \frac{\partial H}{\partial m} = m$, and $\frac{dm}{dt} = -\frac{\partial H}{\partial x_i} = -\nabla_{x_i} U$. Due to the fact that $x_i$ is a Euclidean variable, we use a standard "leap-frog" numerical integration scheme, which approximately conserves the Hamiltonian and results in high acceptance rates.

The computation of the gradient term $\nabla_{x_i} U(x_i)$ requires we compute $d_v \text{Exp}(p, v)$, i.e., the derivative operator (Jacobian matrix) of the exponential map with respect to the initial velocity $v$. To derive this, consider a variation of geodesics $c(s, t) = \text{Exp}(p, su + tv)$, where $u \in T_p M$. The variation $c$ produces a "fan" of geodesics; this is illustrated for a sphere on the left side of Figure 1. Taking the derivative of this variation results in a Jacobi field: $J_v(t) = dc/ds(0, t)$. Finally, this gives an expression for the exponential map derivative as

$$d_v \text{Exp}(p, v)u = J_v(1). \tag{6}$$

For a general manifold, computing the Jacobi field $J_v$ requires solving a second-order ordinary differential equation. However, Jacobi fields can be evaluated in closed-form for the class of manifolds known as *symmetric spaces*. For the sphere and Kendall shape space examples, we provide explicit formulas for these computations in Section 4. For more details on the derivation of the Jacobi field equation and symmetric spaces, see for instance [6].

Now, the gradient with respect to each $x_i$ is

$$\nabla_{x_i} U = x_i - \tau \Lambda W^T \{ d_{z_i} \text{Exp}(\mu, z_i)^\dagger \text{Log}(\text{Exp}(\mu, z_i), y_i) \}, \tag{7}$$

where $\dagger$ represents the adjoint of a linear operator, i.e.

$$\langle d_{z_i} \text{Exp}(\mu, z_i)\hat{u}, \hat{v} \rangle = \langle \hat{u}, d_{z_i} \text{Exp}(\mu, z_i)^\dagger \hat{v} \rangle.$$

### 3.2.2 M-step: Gradient Ascent

In this section, we derive the maximization step for updating the parameters $\theta = (\mu, W, \Lambda, \tau)$ by maximizing the HMC approximation of the $Q$ function in (5). This turns out to be a gradient ascent scheme for all the parameters since there are no closed-form solutions.

**Gradient for $\tau$:** The gradient of the $Q$ function with respect to $\tau$ requires evaulation of the derivative of the normalizing constant in the *Riemannian normal distribution* (2). When $M$ is a symmetric space, this constant does not depend on the mean parameter, $\mu$, because the distribution is invariant to isometrics (see [8] for details). Thus, the normalizing constant can be written as

$$C(\tau) = \int_M \exp\left(-\frac{\tau}{2} d(\mu, y)^2\right) dy.$$

We can rewrite this integral in normal coordinates, which can be thought of as a polar coordinate system in the tangent space, $T_\mu M$. The radial coordinate is defined as $r = d(\mu, y)$, and the remaining $n - 1$ coordinates are parametrized by a unit vector $v$, i.e., a point on the unit sphere $S^{n-1} \subset T_\mu M$. Thus we have the change-of-variables, $\phi(rv) = \text{Exp}(\mu, rv)$. Now the integral for the normalizing constant becomes

$$C(\tau) = \int_{S^{n-1}} \int_0^{R(v)} \exp\left(-\frac{\tau}{2} r^2\right) |\det(d\phi(rv))| dr\, dv, \tag{8}$$

where $R(v)$ is the maximum distance that $\phi(rv)$ is defined. Note that this formula is only valid if $M$ is a complete manifold, which guarantees that normal coordinates are defined everywhere except possibly a set of measure zero on $M$.

The integral in (8) is difficult to compute for general manifolds, due to the presence of the determinant of the Jacobian of $\phi$. However, for symmetric spaces this change-of-variables term has a simple form. If $M$ is a symmetric space, there exists a orthonormal basis $u_1, \ldots, u_n$, with $u_1 = v$, such that

$$|\det(d\phi(rv))| = \prod_{k=2}^n \frac{1}{\sqrt{\kappa_k}} f_k(\sqrt{\kappa_k} r), \tag{9}$$

where $\kappa_k = K(u_1, u_k)$ denotes the sectional curvature, and $f_k$ is defined as

$$f_k(x) = \begin{cases} \sin(x) & \text{if } \kappa_k > 0, \\ \sinh(x) & \text{if } \kappa_k < 0, \\ x & \text{if } \kappa_k = 0. \end{cases}$$

Notice that with this expression for the Jacobian determinant there is no longer a dependence on $v$ inside the integral in (8). Also, if $M$ is simply connected, then $R(v) = R$ does not depend on the direction $v$, and we can write the normalizing constant as

$$C(\tau) = A_{n-1} \int_0^R \exp\left(-\frac{\tau}{2}r^2\right) \prod_{k=2}^{n} \kappa_k^{-1/2} f_k(\sqrt{\kappa_k}r) dr,$$

where $A_{n-1}$ is the surface area of the $n-1$ hypersphere, $S^{n-1}$. The remaining integral is one-dimensional, and can be quickly and accurately approximated by numerical integration. While this formula works only for simply connected symmetric spaces, other symmetric spaces could be handled by lifting to the universal cover, which is simply connected, or by restricting the definition of the Riemannian normal pdf in (2) to have support only up to the injectivity radius, i.e., $R = \min_v R(v)$.

The gradient term for estimating $\tau$ is

$$\nabla_\tau Q = \sum_{i=1}^N \sum_{j=1}^S \frac{1}{C(\tau)} A_{n-1} \int_0^R \frac{r^2}{2} \exp\left(-\frac{\tau}{2}r^2\right) \prod_{k=2}^{n} \kappa_k^{-1/2} f_k(\sqrt{\kappa_k}r) dr - \frac{1}{2} d(\mathrm{Exp}(\mu, z_{ij}), y_i)^2 dr.$$

**Gradient for $\mu$:** From (4) and (5), the gradient term for updating $\mu$ is

$$\nabla_\mu Q = \frac{1}{S} \sum_{i=1}^N \sum_{j=1}^S \tau d_\mu \mathrm{Exp}(\mu, z_{ij})^\dagger \mathrm{Log}\left(\mathrm{Exp}(\mu, z_{ij}), y_i\right).$$

Here the derivative $d_\mu \mathrm{Exp}(\mu, v)$ is with respect to the base point, $\mu$. Similar to before (6), this derivative can be derived from a variation of geodesics: $c(s, t) = \mathrm{Exp}(\mathrm{Exp}(\mu, su), tv(s))$, where $v(s)$ comes from parallel translating $v$ along the geodesic $\mathrm{Exp}(\mu, su)$. Again, the derivative of the exponential map is given by a Jacobi field satisfying $J_\mu(t) = dc/ds(0, t)$, and we have $d_\mu \mathrm{Exp}(\mu, v) = J_\mu(1)$.

**Gradient for $\Lambda$:** For updating $\Lambda$, we take the derivative w.r.t. each $a$th diagonal element $\Lambda^a$ as

$$\frac{\partial Q}{\partial \Lambda^a} = \frac{1}{S} \sum_{i=1}^N \sum_{j=1}^S \tau (W^a x_{ij}^a)^T \{d_{z_{ij}} \mathrm{Exp}(\mu, z_{ij})^\dagger \mathrm{Log}(\mathrm{Exp}(\mu, z_{ij}), y_i)\},$$

where $W^a$ denotes the $a$th column of $W$, and $x_{ij}^a$ is the $a$th component of $x_{ij}$.

**Gradient for $W$:** The gradient w.r.t. $W$ is

$$\nabla_W Q = \frac{1}{S} \sum_{i=1}^N \sum_{j=1}^S \tau d_{z_{ij}} \mathrm{Exp}(\mu, z_{ij})^\dagger \mathrm{Log}(\mathrm{Exp}(\mu, z_{ij}), y_i) x_{ij}^T \Lambda. \tag{10}$$

To preserve the mutual orthogonality constraint on the columns of $W$, we represent $W$ as a point on the Stiefel manifold $V_q(T_\mu M)$, i.e., the space of orthonormal $q$-frames in $T_\mu M$. We project the gradient in (10) onto the tangent space $T_W V_q(T_\mu M)$, and then update $W$ by taking a small step along the geodesic in the projected gradient direction. For details on the geodesic computations for Stiefel manifolds, see [7].

The MCEM algorithm for PPGA is an iterative procedure for finding the subspace spanned by $q$ principal components, shown in Algorithm 1. The computation time per iteration depends on the complexity of exponential map, log map, and Jacobi field which may vary for different manifold. Note the cost of the gradient ascent algorithm also linearly depends on the data size, dimensionality, and the number of samples drawn. An advantage of MCEM is that it can run in parallel for each data point.

---
**Algorithm 1** Monte Carlo Expectation Maximization for Probabilistic Principal Geodesic Analysis
---
    **Input:** Data set $Y$, reduced dimension $q$.
    Initialize $\mu, W, \Lambda, \sigma$.
    **repeat**
       Sample $X$ according to (7),
       Update $\mu, W, \Lambda, \sigma$ by gradient ascent in Section 3.2.2.
    **until** convergence
---

## 4  Experiments

In this section, we demonstrate the effectiveness of PPGA and our ML estimation using both simulated data on the 2D sphere and a real corpus callosum data set. Before presenting the experiments of PPGA, we briefly review the necessary computations for the specific types of manifolds used, including, the Riemannian exponential map, log map, and Jacobi fields.

### 4.1  Simulated Sphere Data

**Sphere geometry overview:** Let $p$ be a point on an $n$-dimensional sphere embedded in $\mathbb{R}^{n+1}$, and let $v$ be a tangent at $p$. The inner product between tangents at a base point $p$ is the usual Euclidean inner product. The exponential map is given by a 2D rotation of $p$ by an angle given by the norm of the tangent, i.e.,

$$\mathrm{Exp}(p, v) = \cos\theta \cdot p + \frac{\sin\theta}{\theta} \cdot v, \quad \theta = \|v\|. \tag{11}$$

The log map between two points $p, q$ on the sphere can be computed by finding the initial velocity of the rotation between the two points. Let $\pi_p(q) = p \cdot \langle p, q \rangle$ denote the projection of the vector $q$ onto $p$. Then,

$$\mathrm{Log}(p, q) = \frac{\theta \cdot (q - \pi_p(q))}{\|q - \pi_p(q)\|}, \quad \theta = \arccos(\langle p, q \rangle). \tag{12}$$

All sectional curvatures for $S^n$ are equal to one. The adjoint derivatives of the exponential map are given by

$$d_p \mathrm{Exp}(p, v)^\dagger w = \cos(\|v\|) w^\perp + w^\top, \qquad d_v \mathrm{Exp}(p, v)^\dagger w = \frac{\sin(\|v\|)}{\|v\|} w^\perp + w^\top,$$

where $w^\perp, w^\top$ denote the components of $w$ that are orthogonal and tangent to $v$, respectively. An illustration of geodesics and the Jacobi fields that give rise to the exponential map derivatives is shown in Figure 1.

**Parameter estimation on the sphere:** Using our generative model for PGA (3), we forward simulated a random sample of 100 data points on the unit sphere $S^2$, with known parameters $\theta = (\mu, W, \Lambda, \tau)$, shown in Table 1. Next, we ran our maximum likelihood estimation procedure to test whether we could recover those parameters. We initialized $\mu$ from a random uniform point on the sphere. We initialized $W$ as a random Gaussian matrix, to which we then applied the Gram-Schmidt algorithm to ensure its columns were orthonormal. Figure 1 compares the ground truth principal geodesics and MLE principal geodesic analysis using our algorithm. A good overlap between the first principal geodesic shows that PPGA recovers the model parameters.

One advantage that our PPGA model has over the least-squares PGA formulation is that the mean point is estimated jointly with the principal geodesics. In the standard PGA algorithm, the mean is estimated first (using geodesic least-squares), then the principal geodesics are estimated second. This does not make a difference in the Euclidean case (principal components must pass through the mean), but it does in the nonlinear case. We compared our model with PGA and standard PCA (in the Euclidean embedding space). The estimation error of principal geodesics turned to be larger in PGA compared to our model. Furthermore, the standard PCA converges to an incorrect solution due to its inappropriate use of a Euclidean metric on Riemannian data. A comparison of the ground truth parameters and these methods is given in Table 1. Note that the noise precision $\tau$ is not a part of either the PGA or PCA models.

| | $\mu$ | w | $\Lambda$ | $\tau$ |
|---|---|---|---|---|
| Ground truth | $(-0.78, 0.48, -0.37)$ | $(-0.59, -0.42, 0.68)$ | 0.40 | 100 |
| PPGA | $(-0.78, 0.48, -0.40)$ | $(-0.59, -0.43, 0.69)$ | 0.41 | 102 |
| PGA | $(-0.79, 0.46, -0.41)$ | $(-0.59, -0.38, 0.70)$ | 0.41 | N/A |
| PCA | $(-0.70, 0.41, -0.46)$ | $(-0.62, -0.37, 0.69)$ | 0.38 | N/A |

Table 1: Comparison between ground truth parameters for the simulated data and the MLE of PPGA, non-probabilistic PGA, and standard PCA.

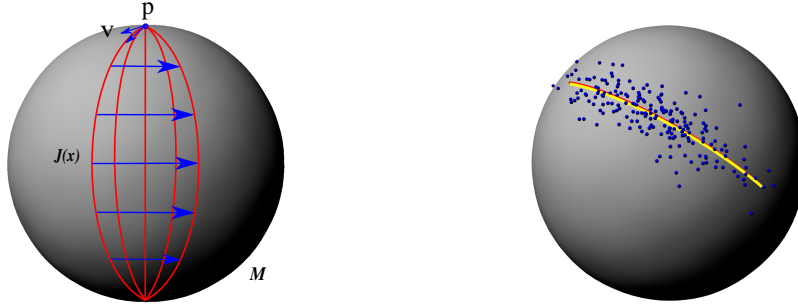

Figure 1: Left: Jacobi fields; Right: the principal geodesic of random generated data on unit sphere. Blue dots: random generated sphere data set. Yellow line: ground truth principal geodesic. Red line: estimated principal geodesic using PPGA.

## 4.2  Shape Analysis of the Corpus Callosum

**Shape space geometry:**  A configuration of $k$ points in the 2D plane is considered as a complex $k$-vector, $z \in \mathbb{C}^k$. Removing translation, by requiring the centroid to be zero, projects this point to the linear complex subspace $V = \{z \in \mathbb{C}^k : \sum z_i = 0\}$, which is equivalent to the space $\mathbb{C}^{k-1}$. Next, points in this subspace are deemed equivalent if they are a rotation and scaling of each other, which can be represented as multiplication by a complex number, $\rho e^{i\theta}$, where $\rho$ is the scaling factor and $\theta$ is the rotation angle. The set of such equivalence classes forms the complex projective space, $\mathbb{C}P^{k-2}$.

We think of a centered shape $p \in V$ as representing the complex line $L_p = \{z \cdot p : z \in \mathbb{C}\backslash\{0\}\}$, i.e., $L_p$ consists of all point configurations with the same shape as $p$. A tangent vector at $L_p \in V$ is a complex vector, $v \in V$, such that $\langle p, v \rangle = 0$. The exponential map is given by rotating (within $V$) the complex line $L_p$ by the initial velocity $v$, that is,

$$\mathrm{Exp}(p, v) = \cos\theta \cdot p + \frac{\|p\| \sin\theta}{\theta} \cdot v, \quad \theta = \|v\|. \tag{13}$$

Likewise, the log map between two shapes $p, q \in V$ is given by finding the initial velocity of the rotation between the two complex lines $L_p$ and $L_q$. Let $\pi_p(q) = p \cdot \langle p, q \rangle / \|p\|^2$ denote the projection of the vector $q$ onto $p$. Then the log map is given by

$$\mathrm{Log}(p, q) = \frac{\theta \cdot (q - \pi_p(q))}{\|q - \pi_p(q)\|}, \quad \theta = \arccos \frac{|\langle p, q \rangle|}{\|p\| \|q\|}. \tag{14}$$

The sectional curvatures of $\mathbb{C}P^{k-2}$, $\kappa_i = K(u_i, v)$, used in (9), can be computed as follows. Let $u_1 = i \cdot v$, where we treat $v$ as a complex vector and $i = \sqrt{-1}$. The remaining $u_2, \ldots, u_n$ can be chosen arbitrarily to be construct an orthonormal frame with $v$ and $u_1$ Then we have $K(u_1, v) = 4$ and $K(u_i, v) = 1$ for $i > 1$. The adjoint derivatives of the exponential map are given by

$$d_p \mathrm{Exp}(p, v)^\dagger w = \cos(\|v\|) w_1^\perp + \cos(2\|v\|) w_2^\perp + w^\top,$$

$$d_v \mathrm{Exp}(p, v)^\dagger w = \frac{\sin(\|v\|)}{\|v\|} w_1^\perp + \frac{\sin(2\|v\|)}{2\|v\|} + w_2^\top,$$

where $w_1^\perp$ denotes the component of $w$ parallel to $u_1$, i.e., $w_1^\perp = \langle w, u_1 \rangle u_1$, $u_2^\top$ denotes the remaining orthogonal component of $w$, and $w^\top$ denotes the component tangent to $v$.

**Shape variability of corpus callosum data:** As a demonstration of PPGA on Kendall shape space, we applied it to corpus callosum shape data derived from the OASIS database (`www.oasis-brains.org`). The data consisted of magnetic resonance images (MRI) from 32 healthy adult subjects. The corpus callosum was segmented in a midsagittal slice using the ITK SNAP program (`www.itksnap.org`). An example of a segmented corpus callosum in an MRI is shown in Figure 2. The boundaries of these segmentations were sampled with 64 points using ShapeWorks (`www.sci.utah.edu/software.html`). This algorithm generates a sampling of a set of shape boundaries while enforcing correspondences between different point models within the population. Figure 2 displays the first two modes of corpus callosum shape variation, generated from the as points along the estimated principal geodesics: $\mathrm{Exp}(\mu, \alpha_i w_i)$, where $\alpha_i = -3\lambda_i, -1.5\lambda_i, 0, 1.5\lambda_i, 3\lambda_i$, for $i = 1, 2$.

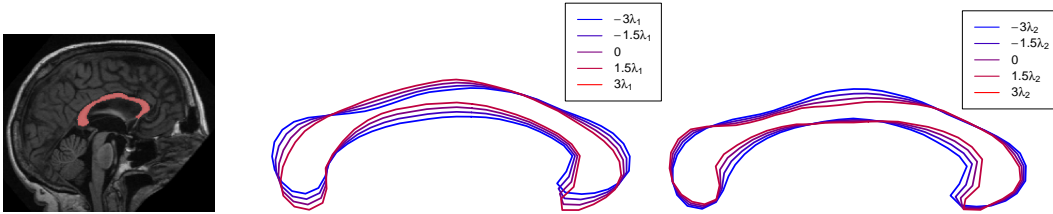

Figure 2: Left: example corpus callosum segmentation from an MRI slice. Middle to right: first and second PGA mode of shape variation with $-3, -1.5, 1.5$, and $3 \times \lambda$.

## 5 Conclusion

We presented a latent variable model of PGA on Riemannian manifolds. We developed a Monte Carlo Expectation Maximization for maximum likelihood estimation of parameters that uses Hamiltonian Monte Carlo to integrate over the posterior distribution of latent variables. This work takes the first step to bring latent variable models to Riemannian manifolds. This opens up several possibilities for new factor analyses on Riemannian manifolds, including a rigorous formulation for mixture models of PGA and automatic dimensionality selection with a Bayesian formulation of PGA.

**Acknowledgments** This work was supported in part by NSF CAREER Grant 1054057.

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
