[Reviews · NeurIPS 2013]

Submitted by Assigned_Reviewer_2

The two main contributions of this paper are: the probabilistic formulation of principal geodesic analysis (PGA) by consideration of a Riemannian-normal noise model; and an exact MCEM scheme for maximum-likelihood estimation of its parameters.

The writing is very clear. The paper feels equally original and incremental. PGA is motivated by the manifold nature of shape and directional data, to name a few. Its probabilistic interpretation, in turn, is motivated by the need to characterize noise in such scenarios, akin to what PPCA (Gaussian), sparse/robust PCA (Laplace, Student's-T) do for Euclidean data. As such, it is an important novel addition to existing analyses of manifold data.

Notes & observations:

- In regards to the HMC step, I'm missing a citation on an existing HMC approach on embedded manifolds: Geodesic Monte Carlo on Embedded Manifolds, Simon Byrne, Mark Girolami, arXiv:1301.6064v1

- Please include some discussion on the computational complexity of the MCEM scheme.

- An algorithm summary at the end of section 3 would improve the reading.


Summary: The authors introduce the first probabilistic interpretation of principal geodesic analysis, that I'm aware of, and an MCEM algorithm for learning.
However the complexity of the algorithm is not discussed and the authors do not seem aware of some relevant prior work (Byrne and Girolami, 2013).

I've read the author's rebuttal.

Submitted by Assigned_Reviewer_5

The paper introduces probabilistic principal component analysis on
Riemannian manifolds, extending earlier non-probabilistic versions to
a probabilistic latent variable model, and derives maximum likelihood
estimation procedures for a broad class of manifolds. The methods are
demonstrated on toy data (maniold is a sphere) and shape analysis on
images.

This is a very interesting advancement, and the paper is well written,
making it reasonably accessible in spite of the difficult topic.

I have a set of interrelated questions; explicating and clarifying
them would clarify the potential impact of the paper to the reader:

- Is essential generality lost by assuming an Euclidean latent space?
Locally on a tangent space it makes sense, and may be practically
necessary, of course.

- It would be good to summarize how much of the analysis depends on
locality approximations, if any. In eqn (7), the integration is only
up to the injectivity radius which is a locality assumption, or is it?

- The analysis of the case studies starts by deriving the Exp maps;
how generally can that be done easily?

- The model reduces to standard PPCA for Euclidean spaces. How about
earlier non-probabilistic versions: does it reduce to them on
Riemannian spaces, under some simplifications?

- It would be great if the model would have been compared to simpler
alternatives in the experiments: non-probabilistic Riemannian PCA, and
simple re-normalized standard PCA.

A minor issue: There are two different notations for the
exponential map.

Quality: As far as I can tell, the paper is technically sound.

Clarity: Very clearly written.

Originality: The probabilistic latent variable treatment is
new. Mixture models have been treated earlier, however, for instance
by Bhattacharya and Dunson, Biometrika 2010, 97:851-865.

Significance: New types of modelling approaches may follow, which take
better into account known constraints in data spaces. Developing them
will require much more special expertise than more common models,
however.

Summary: A very interesting clearly written paper introducing a PCA-type
probabilistic latent variable model operating on Riemannian manifolds.

Submitted by Assigned_Reviewer_6

The paper "Probabilistic Principal Geodesic Analysis" introduces and discusses a probabilistic
formulation of Principal Geodesic Analysis. Principal Geodesic Analysis has been introduced
before as a generalization of standard PCA to non-linear manifolds, i.e., to spaces with
(potentially) other intrinsic geometries than Euclidean spaces. The authors explicitly point
out and discuss earlier PGA results. It is argued that the contribution of the paper is a first
fully probabilistic formulation of PGA and accompanying solutions: In analogy to probabilistic
PCA, probabilistic PGA (PPGA) formulates the problem as a probabilistic generative model.
In this way, the solution can be interpreted as maximum likelihood optimization. The authors
provide an approximate gradient optimization to find the optimal parameters of the PPGA
generative model. As a proof of concept, the numerical results for (A) artificial sphere data
and (B) for shape estimation with corpus callosum data are provided.

Evaluation:
The paper contributes to a very important research direction: statistics for data within
non-Euclidean spaces. A generative formulation of PPCA within a non-Euclidean space
is certainly interesting and potentially of high relevance. I also agree with the authors
that formulations of latent variable models for non-Euclidean spaces is an interesting
future research subject.

The generative formulation of PPGA also makes sense and builds up on previous work
on PGA. The problems I have with the paper are (A) with the formulation of the inference
and learning scheme, and (B) with the experiments.

Formulation of inference and learning:
The paper first makes the impression of providing solutions for general PPGA.
While this may be the case for the probabilistic formulation of PPGA generative model,
the main contribution is rather the inference and learning procedure. The latter is, however,
only really provided for symmetric spaces. This should be very clearly spelled out, in the
abstract and maybe even in the title. The inference and learning procedure, and the
experiments, do rely on symmetric spaces. In this respect, please check formula (4). The
text says the rhs is proportional to the posterior probability but then the normalizer
of equation (2) seems to be missing. The normalizer would be relevant for derivatives w.r.t.
\mu and \tau it seems. The independence of \mu already seems to be used for the gradient
w.r.t. \mu but is only true for symmetric spaces. The paper at least gives the impression
that at this point symmetric spaces are not assumed. For the \tau derivation, symmetric
spaces are explicitly assumed. A lot of more general considerations like the locality of
log map etc could presumably be simplified if the paper focussed on symmetric spaces right
from the start.

Experiments:
The experiments should be clearly labelled as proof of concept. They basically say that
the algorithm seems to work in principle. There is no comparison with other methods.
It would have been interesting to compare with non-probabilistic PGA. As the methods
are essentially equivalent for finding the principle geodesic directions, results should be
similar. PPGA can potentially provide more information (other parameters, maybe a
likelihood estimate) but non of this is shown or discussed. Also potential local optima
are not discussed and other issues about the stability and complexity of the algorithm.

More generally:
Work out and discuss in what situations the approach really has a (potential) advantage
over least square formulations of PGA, or over projection methods such as azimuthal
equidistant projections. When does the generative formulation really pay of? I think
it can.


Other points:

I would cite Roweis for PPGA alongside the other citations

Optimization of the PPCA likelihood is convex, can anything be said
about PPGA optimization (maybe for specific manifold properties).

"requires we compute", "evaulation", "coule", "the as points"



After author response:

The author response has clarified major points. I am also confident that the promised
changes and discussions will improve the manuscript. Especially an earlier mentioning
of the constraint (of the algorithm) to symmetric spaces will make this a clearer
paper. I suggested an explicit mentioning in the abstract because this is an important
point for the paper. The authors promise to point out the constraint in the introduction,
I still think it should be pointed out in the abstract; why not? Symmetric spaces are
still large and interesting and potential later confusions can be removed. Eqn. 4 was
not complete and will be corrected - which will avoid related confusions.

In summary, yes, interesting paper, accept.












Summary: Potentially sound formulation of an important research direction. For symmetric spaces
only (not spelled out initially). Experiments are proof-of-concept.
Author Feedback

Author rebuttal: We would like to thank the reviewers for their thoughtful and helpful suggestions for improvement. Common concerns of all the reviewers are the computational complexity and comparison to other methods, such as non-probabilistic PGA, and standard PCA.

As for the computational complexity, the run time of our algorithm mostly depends on the sampling process in E-step. The computation time per iteration depends on the complexity of exponential map, log map, and Jacobi field which may vary for different manifold. Note the cost of gradient ascent algorithm also linearly on the data size, dimensionality and the number of samples drawn. Since the latent variables for our data set on manifolds live in 1-D dimension, drawing an MCMC sample in each iteration of the EM algorithm is computationally inexpensive and straightforward. Of course, the number of iterations needed for EM to converge is unknown. We will add discussion of the computational complexity to the paper. Another advantage of MCEM is that it can run in parallel for each data point.

We compared our model with non-probabilistic PGA for the examples in the paper, our model outperformed the non-probabilistic PGA for fitting data. In addition, compared with standard PCA (in the Euclidean embedding space), the experiment on high-dimensional sphere data and shape data using our model was more reasonable and accurate. The standard PCA converges to an incorrect solution due to its inappropriate use of a Euclidean metric on Riemannian data. The data error turned to be much larger compared to our model as well as incorrect estimation of principal components.

Specific responses to the reviewers' questions:
Reviewer_2:

According to the reviewer's suggestions, we will include the reference to Byrne et al. and add an algorithm summary in Section 3 if the space is allowed.

Reviewer_5:

Regarding the use of a Euclidean latent space and locality assumptions: The latent variables are naturally coordinates in the tangent space to the mean, which is a Euclidean space. Points on the manifold are generated from these via the exponential map. When a manifold is complete (which is the case for symmetric spaces and many other spaces), any point on the manifold may be generated this way. Thus, there is no loss of generality or locality assumption being made in these cases. For non-complete manifolds, there would be an assumption that data (and the distribution integrated in Eq. 7) are within the injectivity radius from the mean point.

Regarding the Exp map derivations: The Exp map can be formulated on any smooth manifold as the solution to an ordinary differential equation. Computing the Exp map is considerably easier on symmetric spaces (where it has a closed-form solution). On other manifolds it may be more difficult, i.e., requiring numerical integration, but still possible.

We will add the reference to Bhattacharya and Dunson, and we note that their kernel density estimates used isotropic Gaussians on manifolds, which would be different from a mixture of PPGA models.

Reviewer_6:

We agree with the reviewer that our PPGA model definition is general (works on any manifold), but the explicit computations in the inference procedure are only applicable to symmetric spaces. However, our proposed framework should also work on more general spaces as long as the normalizing constant can be computed. We will state this explicitly in the introduction. We will clarify the assumption of independence of the constant wrt \mu (this is true for symmetric spaces, and also for homogeneous spaces, which are more general).

One advantage that our PPGA model has over the least-squares PGA formulation is that the mean point is estimated jointly with the principal geodesics. In the standard PGA algorithm, the mean is estimated first (using geodesic least-squares), then the principal geodesics are estimated second. This does not make a difference in the Euclidean case (PCs must pass through the mean), but it does in the nonlinear case. We can give examples where data can be fit better when jointly estimating mean/PGA than when doing them sequentially.

We agree with the reviewer that there is a typo error in (4), the normalizing constant. We will also cite Roweis et al.

Other minor suggestions of the reviewers will also be carefully incorporated.